# Antibacterial Amphiphilic Copolymers of Dimethylamino Ethyl Methacrylate and Methyl Methacrylate to Control Biofilm Adhesion for Antifouling Applications

**DOI:** 10.3390/polym13020216

**Published:** 2021-01-09

**Authors:** Shehla Mushtaq, Nasir M. Ahmad, Azhar Mahmood, Mudassir Iqbal

**Affiliations:** 1Department of Chemistry, School of Natural Sciences, National University of Sciences and Technology, H-12, Islamabad 44000, Pakistan; shehla.mushtaq@sns.nust.edu.pk (S.M.); dr.azhar@sns.nust.edu.pk (A.M.); mudassir.iqbal@sns.nust.edu.pk (M.I.); 2Polymer Research Lab, School of Chemical and Material Engineering (SCME), National University of Sciences and Technology, H-12, Islamabad 44000, Pakistan

**Keywords:** copolymerization, hydrophilic, antifouling, *E. coli*, *S. aureus*, biofilm

## Abstract

Amphiphilic copolymers are recognized as important biomaterials and used as antibacterial agents due to their effective inhibition of bacterial growth. In current study, the amphiphilic copolymers of P(DMAEMA-co-MMA) were synthesized using free radical polymerization by varying the concentrations of hydrophilic monomer 2-dimethylamino ethylmethacrylate (DMAEMA) and hydrophobic monomer methyl methacrylate (MMA) having PDI value of 1.65–1.93. The DMAEMA monomer, through ternary amine with antibacterial property optimized copolymers, P(DMAEMA-co-MMA), compositions to control biofilm adhesion. Antibacterial activity of synthesized copolymers was elucidated against Gram-positive *Staphylococcus aureus* (ATCC 6538) and Gram-negative *Escherchia coli* (ATCC 8739) by disk diffusion method, and zones of inhibition were measured. The desired composition that was PDM1 copolymer had shown good zones of inhibition i.e., 19 ± 0.33 mm and 20 ± 0.33 mm for *E. coli* and *S. aureus* respectively. The PDM1 and PDM2 have exhibited significant control over bacterial biofilm adhesion as tested by six well plate method. SEM study of bacterial biofilm formation has illustrated that these copolymers act in a similar fashion like cationic biocide. These compositions viz. PDM1 and PDM2, may be useful in development of bioreactors, sensors, surgical equipment and drug delivery devices.

## 1. Introduction

Biofouling is the growth of microbes on exterior of material that initiated by protein adsorption or microorganisms on the surface of substrate that is a ubiquitous challenge for a number of bio medical applications [1,2]. Biofouling also occurs on different prosthetic devices, surgical equipment, protective apparel, sensors, drug delivery devices, contact lenses, medical implants, and bioreactors that causes adverse effects on human health [3,4]. The unregulated attachment of microbes on implant materials surface is an undesirable phenomenon that causes infection and degradation of the function of medical instruments [5,6]. In medical devices antifouling materials are originated by intermolecular interaction between extracellular biomolecules and the designed surfaces [7,8,9]. Most commonly used antimicrobial reagents for bio medical applications can be classified into four specific categories that include organic agents such as formaldehyde and isothiazolones, oxidants that include chlorine and peroxides, electrophilic agents such as mercury, copper and silver, cationic active compounds such as quaternary ammonium and chlorhexidine [10,11]. Use of antimicrobial polymers for biomedical applications has become more significant due to several advantages such as non-volatility, chemically stability, and skin impermeability [3,12]. Different antifouling functional groups such as quaternary ammonium group, flouro group, 2-acrylamido-2-methylpropanesulfonic acid (AMPS) and sulfobetaine offer a way to amend the polymer for certain end applications afterwards [13]. The quaternization of polymeric materials played crucial role in the formulation of new antifouling polymers [14,15]. Polymeric ammonium quaternary compounds have been widely used for antibacterial applications due to the properties such as low volatility, high chemical stability, low toxicity and low skin irritation potential [11,12]. Along with several advantages of quaternary ammonium group to be used as antibacterial agent, there arise some complications during synthesis due to steric and electrostatic effects to obtain a complete quaternization [16,17]. These materials are promising to maintain antibacterial activity and to reduce the risk of toxicity [18]. In one of the study, copolymerization of HEMA with a methacrylic monomer bearing a thiazole side group susceptible to quaternization was carried out, copolymers exhibited significant activity versus Gram-positive (*S. aureus*) and Gram-negative (*P. aeruginosa* and *E. coli*) bacteria [19]. Antimicrobial activity of copolymers increased with increasing of the cationic unit content [19]. Amphiphilic polymers had significant advantage over cationic polymers due to their improved biocidal activity as can be effective against Gram-positive and Gram negative bacteria [14,20]. In another study, synthesis of maleic anhydride and 4-methyl-1-pentene was used to produce amphiphilic copolymers with high antibiotic properties imitating natural antimicrobial peptides [15]. The copolymer obtained was modified by grafting with 3-(DMAPA), which was further improvised to create polycationic copolymers by quaternizing methyl iodide then dodecyl iodide. Antimicrobial properties of the synthesized copolymers have been tested against *E. coli* and *S. aureus*, showed nominal antibacterial activity against Gram-positive bacteria [15]. Lowe et al. copolymerized DMAEMA monomer with different hydrophobic monomers included; octyl, cyclohexyl, butyl and ethyl methacrylate that improved antibacterial activity of materials [21]. DMAEMA monomer was copolymerized with hydrophobic monomer enhanced antibacterial activity that depend upon suitable spacer length of hydrophobic segment [22]. In one of previous work, amphiphilic copolymers showed worthy antibacterial activity against Gram-positive bacteria and poor activity against Gram-negative bacteria. Xu et al. investigated cationic polymers as the main classes of materials against bacteria through the membrane-lysis mechanism. They studied antibacterial effects of linear and cyclic monomers of cyclic poly(2-(dimethylamino)ethyl methacrylate) (PDMAEMA) based copolymers with various components by the intra-chain click cyclization of α-alkyne-ω-azido via atom transfer radical polymerization [23]. In another study of amphiphilic polymers of thiazole ring 2-(2-(4-methylthiazol-5-yl)ethoxy)carbonyl)oxy)ethyl methacrylate monomer (MTZ) and non-hemotoxic poly(ethylene glycol) side chains (poly(ethylene glycol) methyl ether methacrylate (PEGMA) observed that longer hydrophobic chain, octyl were much more hemotoxic than their corresponding butylated copolymers [24].

In this study we report a facile approach to enhance the antimicrobial activity of amphiphilic copolymers by varying concentration of hydrophilic and hydrophobic monomers. These amphiphilic copolymers were synthesized via free radical polymerization with presence of initiator. Desired antimicrobial properties were achieved by t-amine functional group of DMAEMA that endowed low biofilm adhesion. Antibacterial assay was performed against standard Gram-positive *S. aureus* (ATCC 6538), and Gram-negative *E. coli* (ATCC 8739) through disk diffusion method in terms of inhibitory zone diameters (mm). These copolymers showed effective results for Gram-positive and Gram-negative bacteria, especially PDM1 besides PDM2. Amphiphilic copolymers PDM1 and PDM2 had shown low adhesion of microorganisms and mode of action against Gram-positive and Gram-negative bacteria were studied by SEM. PDM1 and PDM2, both compositions are suitable for the design of material with low adhesion of microorganisms needed, especially for antifouling surfaces. Schematic presentation of the synthesized P(DMAEMA-co-MMA) copolymers and the antifouling mechanism is shown in Figure 1.

## 2. Materials and Methodology

### 2.1. Materials

All chemicals were of analytical grade and used in chemical synthesis without further purification. Dimethylamino ethyl methacrylate (DMAEMA, 98%) and Trpytosoy broth (TSB) (Sigma-Aldrich, Humburg, Germany), Methyl methacrylate (MMA, 99%), 2,2-Azobisisobutyronitrile (AIBN, 98%), N,N-Dimethyl formamide (DMF, 99%) (Sigma Aldrich, St. Louis, MO, USA), Ethanol, Paraformaldehyde (PFA) and NaOH were acquired from Sigma Aldrich, Germany while water acquired from 18Ωmill pore RO Plant, was used in reaction. Phospate Buffer Solution (PBS) (VWR, Amersco, Leuven, Belgium), Mueller-Hinton agar (MHA) (Daejung, Shiheung, Korea) and BG11 medium (Scharlau, Barcelona, Spain) were used for algae growth. For the bioassays, *E. coli* (ATTC 8739) and *S. aureus* (ATCC 6538) were employed as representative strains.

### 2.2. Synthesis of Copolymer P(DMAEMA-co-MMA)

Copolymerization between both components like DMAEMA and MMA was performed by free radical polymerization in the various proportions as shown in Table 1 [25]. For synthesis of PDM1 copolymer, DMAEMA (10 g, 63.6 mmol) and MMA (10 g, 99.8 mmol) were dissolved into 200 mL DMF under inert atmosphere with continuous stirring in polymerization reactor (IKA Eurostar200-P4). AIBN (0.2 g, 1.22 mmol) was introduced into the reaction flask, where reaction was permitted to proceed for 5 h with continuous stirring under nitrogen purging through schlenk line at 70 °C temperature. Synthesized copolymers P(DMAEMA-*co*-MMA) were recovered by freeze-drying and yield of synthesized copolymers was 62–65%. Same procedure was repeated for synthesis of PDM2, PDM3 andPDM4, while employing quantities of both monomers mentioned in Table 1.

### 2.3. Characterization

Infrared spectroscopy was performed by Bruker ALPHATIR spectrometer (Germany) at the rate of 20 scans per minute to investigate functional groups of copolymer. The ^1^H NMR spectra was verified by a Bruker Advance 400 spectrometer and operated by 400MHz, with CDCl_3_ (3 mg/mL) solvent. GPC was conducted to determine molecular weight using DMF as the eluent for observing the monomer conversion. DMF-GPC was recorded at a water 1515 system equipped with three HR waters columns (HR4, HR3 and HR1). This system consisted an Isocratic pump and a RI detector. Further calculations with (DMF were performed containing LiBr (0.01 M) as an eluting agent at a run rate of 1.0 mL/min. Different copolymers of different molecular weights were standardized with polystyrene. SEM analysis (JEOLJSM-6490LA) was performed to check normal and distorted bacterial growth on the polymeric materials at 1 µm. Earlier to SEM examination, the specimens were made moisture free and analysis was performed at 10 kV accelerating voltage.

### 2.4. Antibacterial Bioassay

Antibacterial activity against both bacteria *E. coli* and *S. aureus* was performed by disk diffusion method [25,26]. Bacterial cultures had been activated before performing antibacterial activity, and bacteria were streaked at freshly prepared Muller Hinton Agar (MHA) [26]. These agar plates were put into oven at 37 °C for 24 h. Colony from the new development was mixed into saline and optical density (O.D) set 0.5 by using McFarland standard after centrifuging. MHA agar was put into petri dish in sterile conditions by using a Bunsen burner in streamline flow hood. 100 µL from 0.5 O.D culture media was poured into the centre of new MHA plate and streaked by using cotton swab [18,27]. Culture was absorbed on the medium and samples of 8 mm size were placed on a plate, and prepared petri dishes were placed for 24 h at 37 °C. Zones of inhibition were measured around each polymeric sample and these experiments repeated three times [12,13]. These results were presented as a mean ± standard deviation and *T*-test used for determination of statistical importance (* *p* < 0.05, ** *p* < 0.01, *** *p* < 0.001, **** *p* < 0.0001) [28].

### 2.5. Biofilm Formation Test

Biofilm development test was done for *E. coli* and *S. aureus* in six well plates. All antifouling polymers cut into square shape (0.5 × 0.5 cm^2^). In six well plates, two controlled wells contained 2 mL of tryptosoy broth (TSB), two growth wells contained 1 mL of TSB with PMMA polymer and two treated wells contained 1 mL of TSB with antifouling copolymers [20,29]. Six-well plates of four copolymers were completely wrapped by food packing sheet and incubated at 37 °C for 24 h. Afterward plates were removed from incubator and bring into laminar flow hood. Samples were washed carefully with phosphate buffer solution (PBS) (1/100 mL) to remove unattached cells and subjected them to SEM fixation. 100–200 µL of 4% para formaldehyde (PFA) was placed on washed samples and allowed to dry for 30–40 min [30,31]. Samples were washed with 25%, 50%, 70% and 100% concentrations of ethanol. After drying, samples were preserved at −4 °C until subjected to SEM analysis [32].

## 3. Results and Discussion

Copolymers of both monomers MMA and DMAEMA are shown in Figure 2. P(DMAEMA-co-MMA) was made by free radical polymerization, using AIBN initiator and DMF solvent at 70 °C under inert medium [33,34,35]. Chemical reaction of both monomers MMA and DMAEMA is shown in Figure 3. Different copolymers were synthesized by varying concentration of both monomers and characterize by FTIR, ^1^HNMR and GPC.

### 3.1. FTIR Analysis

PMMA and PDMAEMA homopolymers are characterized by FTIR, as shown in Figure 4a. Here spectrum for PMMA showed absorption bands at 2997 cm^−1^ due to the stretching vibrations of –CH_2_– and –CH_3_ groups [32]. In PMMA spectrum absorption band at 1730 cm^−1^ exhibited stretching vibration of C = O group of ester [32]. In PDMAEMA spectrum band at 1730 cm^−1^ showed C = O group and band at 2842 cm^−1^ attributed to C-H stretching vibration of N(CH_3_)_2_ moieties [33,34]. In both acrylate homopolymers band at 1150–1250 cm^−1^ showed stretching vibration of C-O-C [35]. Copolymerization of MMA and DMAEMA confirmed and showed in Figure 4b. These copolymers from PDM1 to PDM4 contained the characteristic bands of DMAEMA and MMA. In each spectrum, band at 1020 cm^−1^ for C–N stretching vibration of tertiary amine confirms presence of DMAEMA segment into copolymers. Bands at 1730 cm^−1^ for C = O of ester corresponded to MMA moiety in the copolymers [35]. In copolymers PDM1 to PDM4 concentration of DMAEMA decreases and band intensity of C-N group also reduced. On the other hand, concentration of MMA increase from PDM1 to PDM4 hence signal intensity at 1730 cm^−1^ was enhanced [36]. In both Figure 4a,b bands appeared at 1450 cm^−1^ due to bending vibration of –CH2 group. DMAEMA is hydrophilic in nature and it absorb moisture and band due to O-H group at 3000–3500 cm^−1^ in Figure 2a.

All copolymers showed characteristic band for C-O-C at 1150–1250 cm^−1^ and a broad band at 3300–3400 cm^−1^ due to -OH group was clearly observed in PDM1 to PDM3 due to the moisture absorption by DMAEMA. In PDM4, –OH band disappeared because of very low concentration of DMAEMA and higher content of MMA.

### 3.2. ^1^HNMR Analysis

Chemical structure of amphiphilic copolymers PDM1 have been shown in Figure 5. The ^1^HNMR spectrum of –OCH_2_ of PDMAEMA exhibited a peak at 4.1 (c) ppm. The ^1^HNMR (h) peak at 3.6 ppm for –OCH3 that corresponds for PMMA [37]. The ^1^HNMR showed that molar ratio of PDMAEMA and PMMA was 1:1, which was determined using the integration of (c) and (h) peaks, which was equivalent to 2 and 3 protons respectively. In PDMAEMA segment, dimethylamino groups showed two sharp peaks at 2.26 (e) ppm for two protons of methyl group and 2.54 (d) ppm for six protons of amino group. Magnetic resonances at 1.0 ppm and 1.33 ppm are associated with the methyl protons of main chains while two signals around 1.8 ppm are assigned to the methyl group [38]. These ^1^HNMR results have confirmed that copolymerization was successfully executed.

### 3.3. GPC Results

The molecular weight of synthesized copolymers with changing concentration of both monomers confirm the controlled feature of polymers number average molecular weights (Mn, GPC) of polymers, as shown in Figure 6 [22]. Random copolymer synthesis and monomer conversion was resulted in high molecular weight polymers [39]. Polydispersity index (PDI), Mw/Mn values characterize samples ranged from 1.65 to 1.93 for free radical polymerization [40]. The symmetrical nature of the GPC curves of four copolymers the inexistence of an irreversible termination of both monomers DMAEMA and MMA [36]. Traces of symmetric GPC distribution showed the uniformity of the copolymers via free radical polymerization. [30]. These unimodal curves showed that the polymerization was completed successfully and that there was no unreacted monomers in the reaction product.

### 3.4. Antibacterial Bioassay

Amphiphilic copolymer of P(DMAEMA-*co*-MMA) are partially soluble in water, PDMAEMA is water soluble and PMMA is water insoluble due to hydrophilic and hydrophobic nature [41]. Antibacterial action was tested by disk diffusion method (DD) and zone of inhibitions were measured against both type of bacteria [42]. Bacterial bond of *E. coli* and *S. aureus* on surface of polymers did not significantly depend upon the molecular weight of polymers, but it dependent on the elementary process of bactericidal action of polymers [12]. PMMA has hydrophobic nature and not antibacterial action, while its copolymers with DMAEMA presented antimicrobial activity against *S. aureus* and *E. coli* [12]. These amphiphilic copolymers PDM1 to PDM4 have shown different zones of inhibitions alongside Gram-positive and Gram-negative bacteria in Figure 7a,b. These amphiphilic copolymers played profound effect on the antibacterial activity [43]. PDM1 has more concentration of DMAEMA monomer; hence demonstrated higher activity against bacteria because it had higher charge density due to amino groups [44]. Furthermore, concluding the biocidal efficacy of those copolymers for Gram-positive bacteria is greater than the Gram-negative, which is also consistent with the outcome stated by Ignatova et al. [45]. Since, the Gram-positive bacteria have cell wall that made up of only peptidoglycan the diffusion for the cationic polyelectrolytes with hydrophobic group are easier [43,46]. For Gram-negative bacteria (*E. coli*) it is however more complicated to diffuse over the cell wall while cells are surrounded by another outside membrane [43,46,47].

### 3.5. Biofilm Adhesion Studies by SEM

Antibacterial activities of all copolymers with different concentrations of DMAEMA were assessed against Gram-negative bacteria and Gram-positive bacteria which commonly cause biofilm on materials [11]. Biofouling resistance of synthesized copolymer materials against *S. aureus* and *E. coli* biofilm formation was studied by six well plate method for incubation time of 24 h. Bacterial cell had negative charge on the surface so easily attached on the cationic surface while its antibacterial activity was enhanced by molecular weight [43,48]. Antibacterial activity also influenced by the spacer length due to conformation charge density on the polymers [13]. As well as DMAEMA monomer had positive charge because ammonia group copolymerized and MMA had no charge with hydrophobic in nature [11]. Adhesion of Gram-positive bacteria *S. aureus* on the surface of copolymers shown in Figure 8 [6]. The increase in antibacterial activity increase with charge density by polymerization and assumed more adhesion due to negative charged bacterial cell surface, increase dispersal through the cell wall. These cationic polymers fix to the cytoplasmic membrane, interruption the cytoplasmic membrane, discharge of intracellular elements and bacterial cells mortality [6,49,50].

These amphiphilic polymers destabilize the surface of *E. coli* by interchange with cations of materials that cause rapture of cells as shown in Figure 9 [6]. Copolymer antibacterial action mechanism occurred by (a) bacterial cell surface adsorption, (b) cell wall diffusion, (c) cytoplasmic membrane adsorption, (d) cytoplasmic disruption, (e) leakage of cellular components, and (f) cell death [6,16]. Here more positive charge present on polymers PDM1 and PDM2 that caused interactions between polymers and bacteria. So it is a critical factor, and further action disrupted the cell wall, fluid leaked, cell raptured and death occurred [47,51]. The tertiary amine present in DMAEMA caused disruption of the bacterial cell wall.

## 4. Conclusions

Various copolymers (PDM1, PDM2, PDM3 and PDM4) were successfully prepared by free radical polymerization in the presence of AIBN initiator by employing different ratios of DMAEMA and MMA monomers. These copolymers were characterized by FTIR, ^1^HNMR, and GPC, which exhibited high antimicrobial activities against *E. coli* and *S. aureus.* Bare PMMA showed no antibacterial activity. Copolymer of P(DMAEMA-co-MMA), PDM1 showed maximum biocidal activity with an inhibition zone of 19 mm and 20 mm against *E. coli* and *S. aureus* respectively. In these copolymers, PDM1 and PDM2 had a higher concentration of DMAEMA; therefore, it showed greater antibacterial activities as compared to PDM3 and PDM4. This greater activity is attributed to the presence of amine groups along the chain length of the DMAEMA segment. Adhesion of microorganisms on the surface and biofilm formation decreased with an increase in the molar ratio of DMAEMA due to the presence of positive charges responsible for biocidal action. In SEM analysis of biofilm, the control and rupture of *E. coli* cell membrane was observed in PDM1 and cell disruption of *S. aureus* was observed in PDM2. Thus, PDM1 and PDM2 copolymers are potential candidates for antifouling applications with controlled biofilm formation.

## Figures and Tables

**Figure 1 polymers-13-00216-f001:**
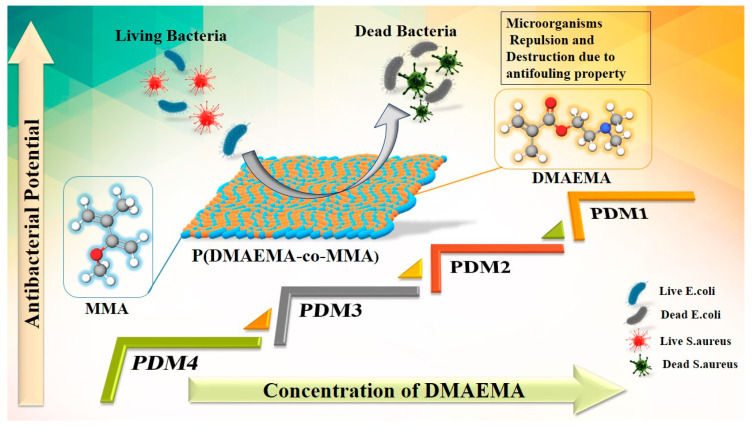
Schematic illustration of copolymers P(DMAEMA and MMA),PDM1, PDM2, PDM3 and PDM4 with varying concentration of DMAEMA that control bacterial adhesion.

**Figure 2 polymers-13-00216-f002:**
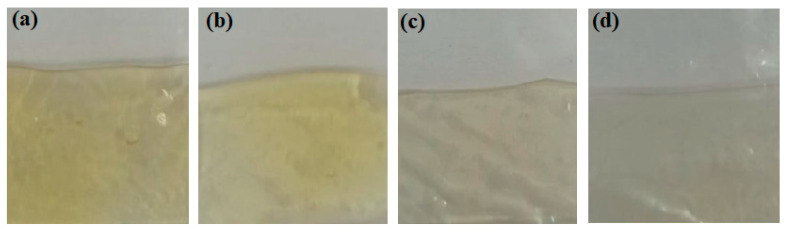
Copolymers of P(DMAEMA-co-MMA) (**a**) PDM1; (**b**) PDM2; (**c**) PDM3 and (**d**) PDM4.

**Figure 3 polymers-13-00216-f003:**
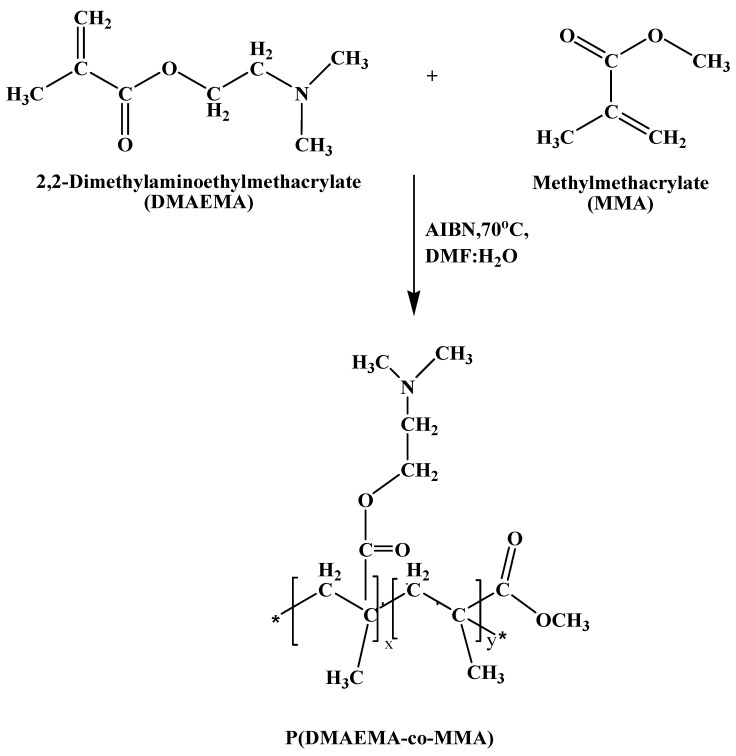
Schematic view of copolymerization reaction between both monomers DMAEMA and MMA.

**Figure 4 polymers-13-00216-f004:**
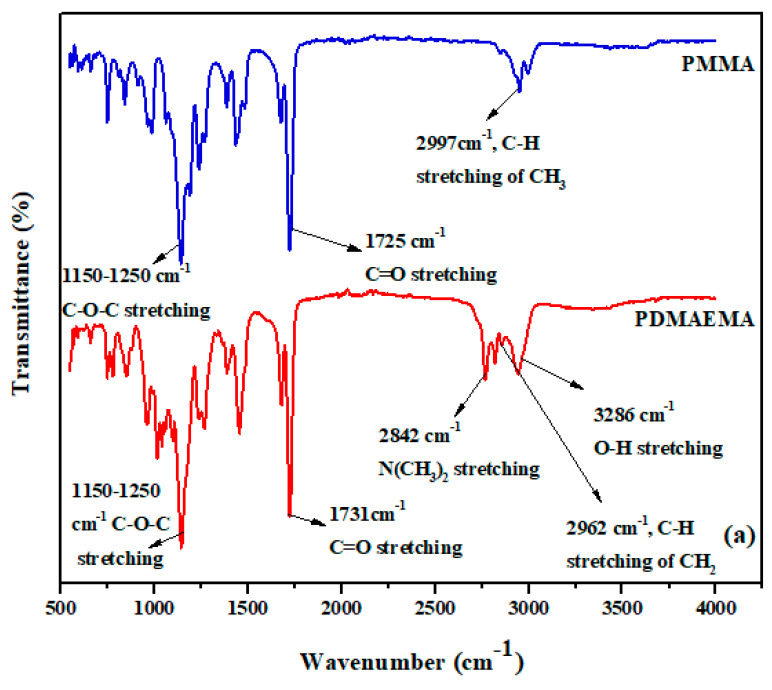
FTIR spectra of: (**a**) Homopolymers PDMAEMA and PMMA; (**b**) Copolymers of P(DMAEMA-co-MMA) PDM1, PDM2, PDM3 and PDM4.

**Figure 5 polymers-13-00216-f005:**
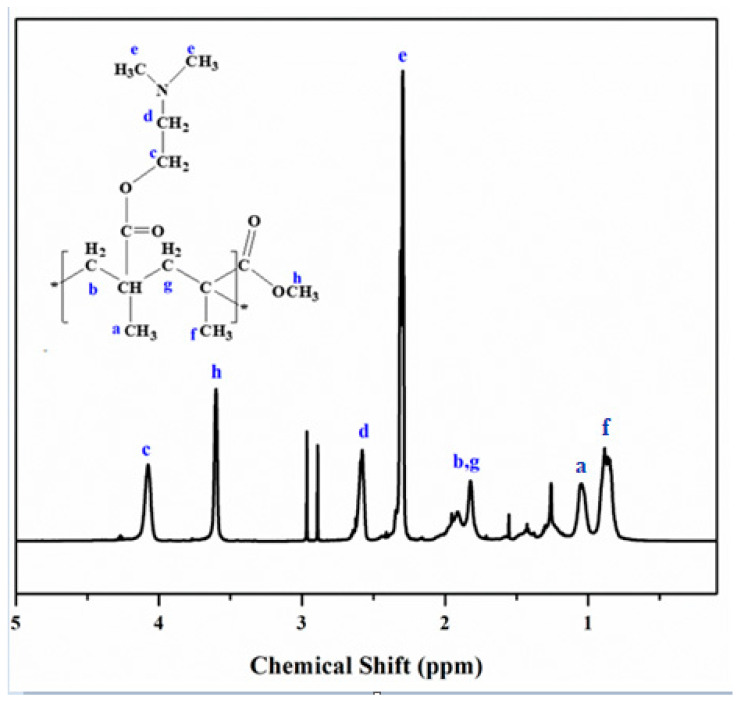
^1^HNMR spectrum of PDM1, P(DMAEMA-*co*-MMA) copolymer.

**Figure 6 polymers-13-00216-f006:**
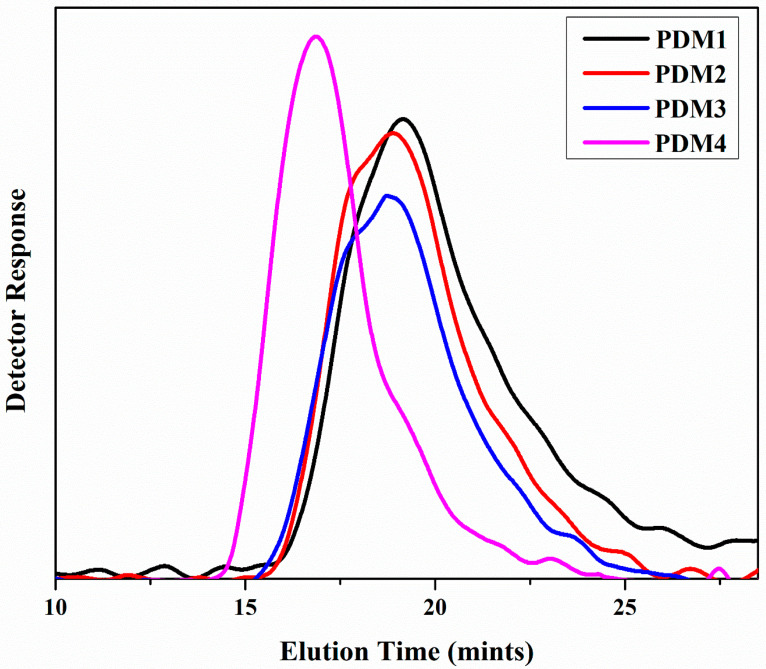
Gel Permeation Chromatography (GPC) analysis of (**a**) PDM1, (**b**) PDM2, (**c**) PDM3 and (**d**) PDM4.

**Figure 7 polymers-13-00216-f007:**
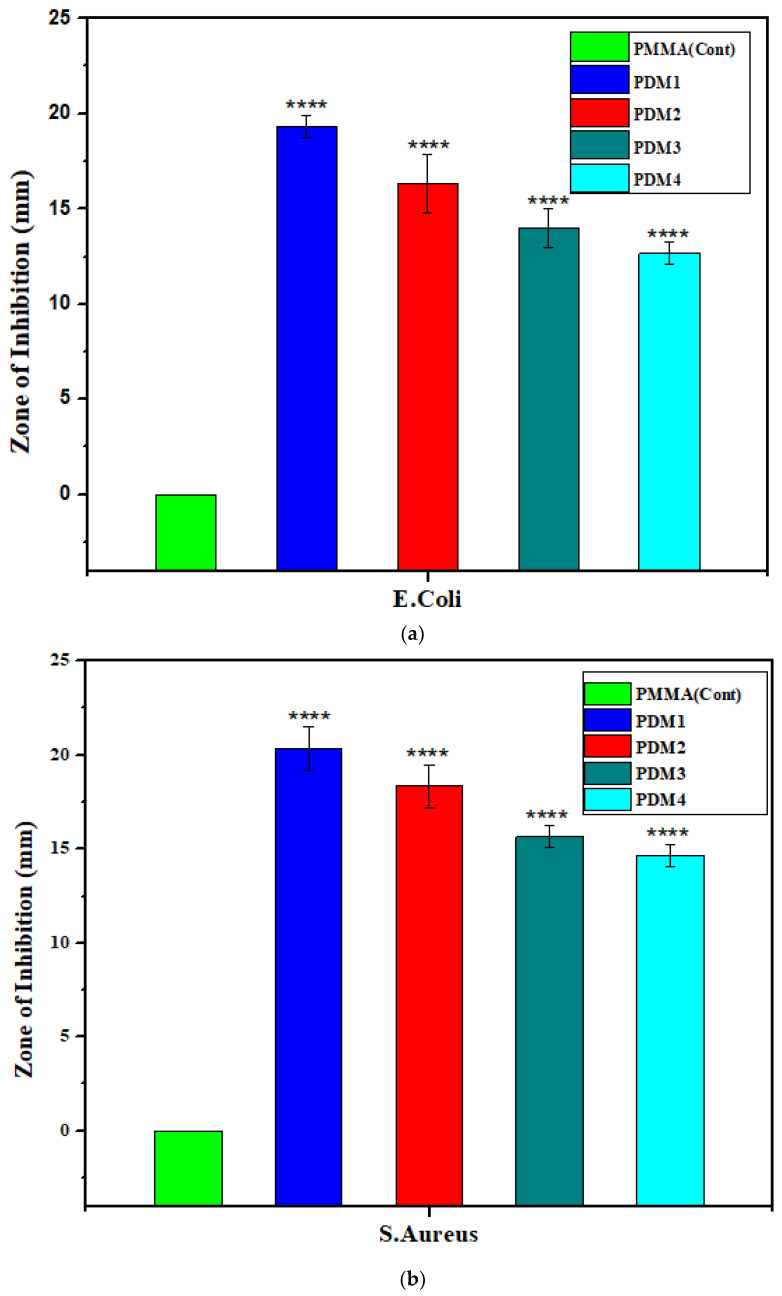
Zone of inhibition of copolymer, PDM1, PDM2, PDM3 and PDM4 with PMMA as a control after t test (**** *p* < 0.0001): (**a**) against *E. coli*; (**b**) against *S. aureus.*

**Figure 8 polymers-13-00216-f008:**
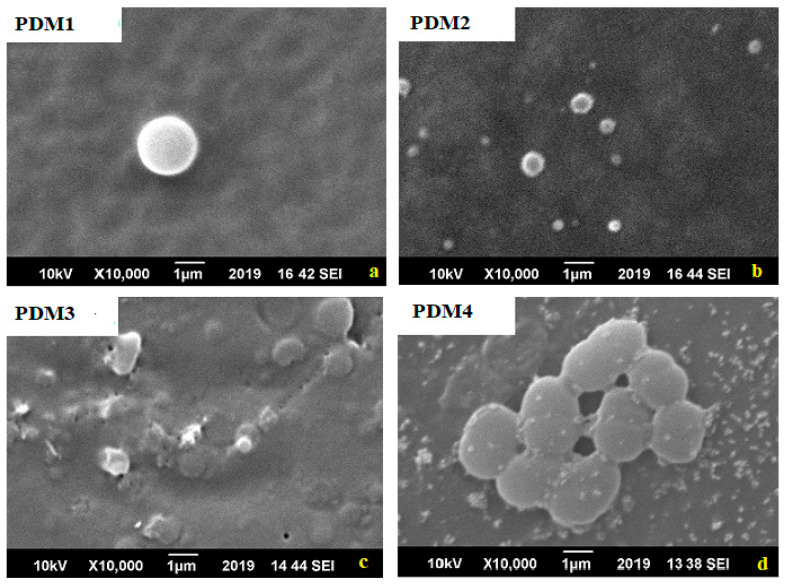
Scanning electron microscopy images for *S. aureus* that shows biofilm formation trend in copolymers: (**a**) PDM1; (**b**) PDM2; (**c**) PDM3; (**d**) PDM4.

**Figure 9 polymers-13-00216-f009:**
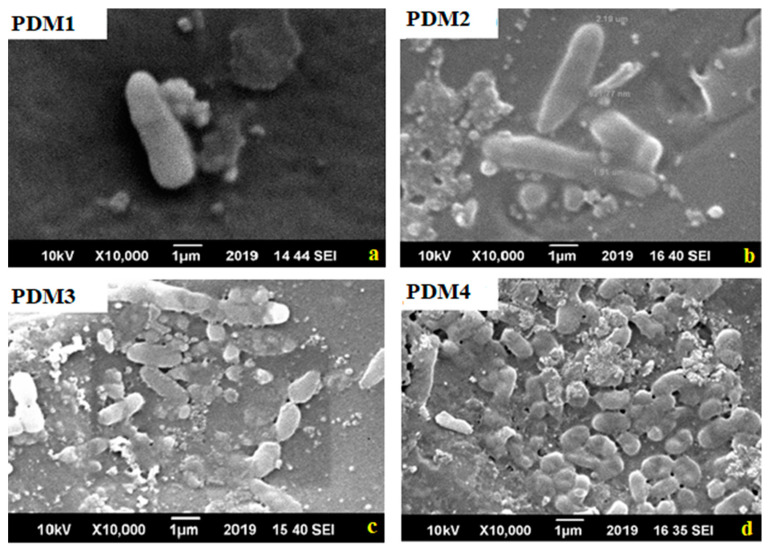
Scanning electron microscopy images for *E. coli* that shows biofilm formation trend in copolymers: (**a**) PDM1; (**b**) PDM2; (**c**) PDM3; (**d**) PDM4.

**Table 1 polymers-13-00216-t001:** Concentration, molar %, average molecular mass (Mn) and polydispersity index of copolymers P(DMAEMA-co-MMA) samples PDM1, PDM2, PDM3 and PDM4 with different molar concentration of DMAEMA and MMA.

Samples	Conc. (mmol)		Mol % (^1^HNMR)		P(DMAEMA-co-MMA)	
	DMAEMA	MMA	DMAEMA	MMA	Mn (g/mol)	PDI
PDM 1	63.6	99.8	44	56	56562	1.75
PDM 2	50.8	119.8	29	71	55865	1.65
PDM 3	38.2	139.8	20	80	54507	1.93
PDM 4	25.4	159.8	14	86	194617	1.88

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
