# Peer review of "Antibacterial Amphiphilic Copolymers of Dimethylamino Ethyl Methacrylate and Methyl Methacrylate to Control Biofilm Adhesion for Antifouling Applications"

_polymers, 2021, doi:10.3390/polym13020216_

Round 1

Reviewer 1 Report

In this study, the authors prepared and characterized a series of amphiphilic copolymers of P(DMAEMA-co-MMA) for antibacterial activity study. Although he results support the main objective of this study, the necessary novelty was not well addressed. List of critical comments are shown as below.

  1. The author should provide the NMR of PDM2-4 for comparison. Based on integration in the NMR, the real Mol% of each polymer can be calculated instead of theoretical number in Table 1.
  2. The author claimed that “more amino groups and positive charges that caused biocidal action”. Why not directly use homopolymer PDMAEMA for this study. It is believed that PDMAEMA would have the best antibacterial activity. Homopolymer PDMAEMA and PMMA should also been studied as control groups.

Author Response

Reviewer 1 Comments

In this study, the authors prepared and characterized a series of amphiphilic copolymers of P(DMAEMA-co-MMA) for antibacterial activity study. Although he results support the main objective of this study, the necessary novelty was not well addressed. List of critical comments are shown as below.

Comment 1. The author should provide the NMR of PDM2-4 for comparison. Based on integration in the NMR, the real Mol% of each polymer can be calculated instead of theoretical number in Table 1.

Reply: NMR of all copolymers of PDM1, PDM2, PDM3 and PDM4 is provided below and Mol% from 1HNMR of all copolymers calculated and given in the table 1. The 1H NMR spectra below shows that ratio of MMA increases from PDM1- PDM4 respectively.

Samples

   Conc.  (mmol)

  Mol % (1HNMR)

P(DMAEMA-co-MMA)

DMAEMA

MMA

DMAEMA

MMA

Mn (g/mol)

PDI

PDM 1

6.36

9.98

40

60

56562

1.75

PDM 2

5.08

11.98

29

71

55865

1.65

PDM 3

3.82

13.98

20

80

54507

1.93

PDM 4

2.54

15.98

13

87

194617

1.88

Comment 2. The author claimed that “more amino groups and positive charges that caused biocidal action”. Why not directly use homopolymer PDMAEMA for this study. It is believed that PDMAEMA would have the best antibacterial activity. Homopolymer PDMAEMA and PMMA should also been studied as control groups.

Reply: Yes, we claimed that more amino groups and more positive charges that caused biocidal action. PDMAEMA, homopolymer is fully water soluble and already studied its antibacterial properties [1]. Main purpose was here to attain antibacterial property by introducing hydrophobic segment (PMMA) with hydrophilic segment PDMAEMA that enhance its antimicrobial activity and make insoluble in water in form of copolymers P(DMAEMA-co-MMA). While PMMA has no antibacterial activity that is reported in our previous work [2]. These copolymers are especially synthesized for antifouling applications. Novelty of this research work is to study the varying concentration effects of DMAEMA and MMA on antibacterial activity.

References

  1. Muñoz-Bonilla A, Fernández-García M. Polymeric materials with antimicrobial activity. Progress in Polymer Science. 2012 Feb 1; 37(2):281-339.
  2. Shehla Mushtaq, Nasir M. Ahmad, Habib Nasir, Azhar Mahmood, Hussnain A. Janjua, "Transpicuous-Cum-Fouling Resistant Copolymers of 3-Sulfopropyl Methacrylate and Methyl Methacrylate for Optronics Applications in Aquatic Medium and Healthcare", Advances in Polymer Technology, vol. 2020, Article ID 5392074, 11 pages, 2020. https://doi.org/10.1155/2020/5392074

Reviewer 2 Report

The manuscript presents the results from the preparation and characterisation of a small family of  copolymers derived from MMA and DMAEMA via radical polymerisation and evaluation of their antibacterial properties and biofilm formation.

The experimental design is based on known and widely investigated free radical polymerisation, the copolymers displayed the expected behaviour. The antibacterial properties of PDMAEMA and copolymers containing DMAEMA units have been studied intensively. The presented investigation does not reveal some specific observations or deeper insight on copolymer interactions with the bacteria or other related issues than the already reported in the scientific literature.

Further, there are many unclear statements and sentences that need revision. I would give some examples but the whole manuscript should be carefully revised. I will also provide additional comments with focus on the synthetic part of the work and hope it will help the authors to improve the quality of the work.

Starting with the title: Antibacterial Biofilm ….. sounds contradictive

Lines 81-82: The sentence needs revision: Schematic of the synthesized copolymers and antifouling mechanism of P(DMAEMA-co-MMA), PDM1 to PDM4 is shown in Figure 1. Probably the authors mean that schematic presentation of the synthesized P(DMAEMA-co-MMA) copolymers and the antifouling mechanism is shown in Figure 1.

According the section 2.2. the copolymerisation was performed in DMF and the obtained products were isolated by freeze-drying. No other purification procedure was described, i.e. precipitation or dialysis. Generally freeze-drying from DMF solution is not recommended because the solvent is aggressive to the equipment. However, purification step is needed to remove the unreacted monomers not only the solvent. Otherwise, it should be accepted that removal of the solvent and unreacted monomers was achieved by freeze-drying. Comments on that issue are not provided in the manuscript.

Other questions to be addresses:

How much were the yields of the copolymer products?

How were the copolymer samples prepared for the bioassays?

Figure 2 is too large.

The assignment of the characteristic absorbances for the PMMA, PDMAEMA and the copolymers is not complete. Both homopolymers are methacrylates and display similar bands, for example the C=O stretching which is commented as a characteristic vibration for PMMA though presented spectra in Figure 3. On the other hand, there are absorbances in the copolymer spectra between 1500 cm-1 and 1700 cm-1 which are not present in the spectra of the homopolymers and not commented.  They could be due to residual monomers or solvent and therefore purification step is recommended.

The interpretation of the 1H NMR spectrum should also be carefully revised. There are errors that could be typing but anyway they should be excluded. For example: ester methylene (–OCH3),

two protons of methyl group, six protons of amino group. There are indications that residual solvent (signals at ∼3 ppm) and probably some unreacted monomers (signals at ∼2 ppm) are present in the products.

The data about the composition of the copolymers calculated from the NMR data to be added in Table 1.

The whole section 3.3. GPC results should be reconsidered. There is absolutely unclear statements: ”In synthesis of copolymer, monomers conversions resulted in a distribution of segment and formation of high molecular weight from bimolecular weighed distribution (MWD)”or “There was slight molecular weight distribution Mw/Mn, <1.4 of all copolymers [31].”

Section 3.4. Antibacterial Bioassay. Similarly, this section also contains unclear statements, for example “These polymethacrylamide random copolymers with different  concentration of MMA and DMAEMA, have bearing protonated primary amine groups.”

In conclusion, I cannot recommend the manuscript for publication. The authors should revise the text and also if possible to perform additional biological experiments and discuss more deeply the behaviour of the copolymers with different composition and to submit the work to more specialized journal in microbiological problems.

Author Response

Reviewer 2 Comments

The manuscript presents the results from the preparation and characterisation of a small family of copolymers derived from MMA and DMAEMA via radical polymerisation and evaluation of their antibacterial properties and biofilm formation.

Comment 1. The experimental design is based on known and widely investigated free radical polymerization, the copolymers displayed the expected behavior. The antibacterial properties of PDMAEMA and copolymers containing DMAEMA units have been studied intensively. The presented investigation does not reveal some specific observations or deeper insight on copolymer interactions with the bacteria or other related issues than the already reported in the scientific literature.

Reply: Antibacterial properties of copolymers are studied by disk diffusion method and mechanism of action was clearly described in paper in lines (230-237).

Antibacterial action was tested by disk diffusion method (DD) and zone of inhibitions were measured against both type of bacteria [43]. Bacterial bond of E. coli and S. aureus on surface of polymers did not significantly depend upon the molecular weight of polymers, it dependent on the elementary process of bactericidal action of polymers [14]. PMMA has hydrophobic nature and not antibacterial action while its copolymers with DMAEMA presented antimicrobial activity against S. aureus and E. coli [14]. These amphiphilic copolymers PDM1 to PDM4 have shown different zone of inhibitions alongside Gram-positive and Gram-negative bacteria in Figure 7a and 7b. These amphiphilic copolymers played profound effect.   In this research article clearly describe composition effects of copolymers on antibacterial activity. Further biofilm formation on the surface of polymers was studied by six well plate method. After fixation of biofilm, it was further studied by scanning electron microscopy. After SEM mechanism of killing of microorganism is cleared as described in lines (250-276).

Antibacterial activities of all copolymers with different concentrations of DMAEMA were assessed against Gram-negative bacteria and Gram-positive bacteria which commonly cause biofilm on materials [18]. Biofouling resistance of synthesized copolymer materials against S. aureus and E. coli biofilm formation was studied by six well plate method for incubation time of 24 hours. Bacterial cell had negative charge on the surface so easily attached on the cationic surface while its antibacterial activity enhanced by molecular weight [44,49]. Antibacterial activity also influenced by the spacer length due to conformation as well as charge density on polymers [15]. DMAEMA monomer had positive charge because ammonia group copolymerized and MMA had no charge with hydrophobic in nature [18]. Adhesion of Gram-positive bacteria S. aureus on the surface of copolymers shown in Figure 8 [6].  The increase in antibacterial activity increase with charge density by polymerization and assumed more adhesion due to negative charged bacterial cell surface, increase dispersal through the cell wall. These cationic polymers fix to the cytoplasmic membrane, interruption the cytoplasmic membrane, discharge of intracellular elements and bacterial cells mortality [6,50,51]. These amphiphilic polymers destabilize the surface of E. coli by interchange with cations of materials that cause rapture of cells as shown in Figure 9 [6]. Copolymer antibacterial action mechanism occurred by (a) bacterial cell surface adsorption, (b) cell wall diffusion, (c) cytoplasmic membrane adsorption, (d) cytoplasmic disruption, (e)  leakage of cellular components and (f) cell mortality (f) cell death [6,19]. Here more positive charge present on polymers PDM1 and PDM2 that caused interactions between polymers and bacteria so it’s a critical factor and further action disrupted the cell wall, fluid leaked, cell raptured and death occurred  [52]. Tertiary amine present in DMAEMA caused disruption of bacterial cell wall.

Comment 2. Further, there are many unclear statements and sentences that need revision. I would give some examples but the whole manuscript should be carefully revised. I will also provide additional comments with focus on the synthetic part of the work and hope it will help the authors to improve the quality of the work.

Starting with the title: Antibacterial Biofilm ….. Sounds contradictive

Reply: This contradiction is solved in title now.

Antibacterial Amphiphilic Copolymers of Dimethylamino Ethyl Methacrylate and Methyl Methacrylate to Control Biofilm Adhesion in Antifouling Applications.

Comment 3. Lines 81-82, The sentence needs revision: Schematic of the synthesized copolymers and antifouling mechanism of P(DMAEMA-co-MMA), PDM1 to PDM4 is shown in Figure 1. Probably the authors mean that schematic presentation of the synthesized P(DMAEMA-co-MMA) copolymers and the antifouling mechanism is shown in Figure 1.

Reply: Line 81-82 sentence was revised according to reviewer suggestion and now sentence is

Schematic presentation of the synthesized P(DMAEMA-co-MMA) copolymers and the antifouling mechanism is shown in Figure 1

Comment 4. According the section 2.2. The copolymerization was performed in DMF and the obtained products were isolated by freeze-drying. No other purification procedure was described, i.e. precipitation or dialysis. Generally freeze-drying from DMF solution is not recommended because the solvent is aggressive to the equipment. However, purification step is needed to remove the unreacted monomers not only the solvent. Otherwise, it should be accepted that removal of the solvent and unreacted monomers was achieved by freeze-drying. Comments on that issue are not provided in the manuscript.

Reply: As with other higher molecular weight, water-soluble polymers, the mode of polymer purification and isolation can also have a profound effect on polymer solution properties. This effect is especially pronounced at high [M /[I]1/2 ratios or higher molecular weight polymers, suggesting the loss or breakdown of some polymer upon precipitation redissolution  or the removal of low molecular weight impurities via dialysis. So here used freeze drying over precipitation and dialysis.

Reference: D.N. Schulz, J.J. Kaladas, J.J. Maurer, J. Bock, S.J. Pace, W.W. Schulz,Copolymers of acrylamide and surfactant macromonomers: synthesis and solution properties, Polymer, Volume 28, Issue 12, 1987,Pages 2110-2115.

Comment 5. How much were the yields of the copolymer products?

Reply: Various copolymer were synthesized with yields were 62-65 %. This is included into the experimental section in lines (115-116).

Comment 6. How were the copolymer samples prepared for the bioassays?

Reply: Copolymer samples need no preparation or pretreatment was needed just washing with autoclaved distilled water and samples cut of required sizes. These samples of specific sizes further used after washing in disk diffusion and biofilm formation test are performed to check antifouling activity.

Comment 7. Figure 2 is too large. The assignment of the characteristic absorbances for the PMMA, PDMAEMA and the copolymers is not complete. Both homopolymers are methacrylates and display similar bands, for example the C=O stretching which is commented as a characteristic vibration for PMMA though presented spectra in Figure 3. On the other hand, there are absorbances in the copolymer spectra between 1500 cm-1 and 1700 cm-1 which are not present in the spectra of the homopolymers and not commented.  They could be due to residual monomers or solvent and therefore purification step is recommended.

Reply: In Figure 3a PMMA and MMA showed different bands that are represent in 2a and there is similarity because both homopolymers are acrylates and maximum groups are similar. While in PDMAEMA –N(CH3)2 is different group as compared to MMA and its peak is only visible in PDMAEMA spectrum. In both Figures 3a of homopolymers and 3b of copolymers, bands appeared at 1450 cm-1 due to bending vibration of –CH2 group. All copolymers showed characteristic band for C-O-C at 1150-1250cm-1. Now in both Figures 2a and 2b all peaks are mentioned and commented in manuscript. Lines (183-185) & (192-193).

Comment 8. The interpretation of the 1H NMR spectrum should also be carefully revised. There are errors that could be typing but anyway they should be excluded. For example: ester methylene (–OCH3), two protons of methyl group, six protons of amino group. There are indications that residual solvent (signals at ∼3 ppm) and probably some unreacted monomers (signals at ∼2 ppm) are present in the products. The data about the composition of the copolymers calculated from the NMR data to be added in Table 1.

Reply: Data of all compositions calculated from 1HNMR was added in table 1.

Samples

   Conc.  (mmol)

  Mol % (1HNMR)

P(DMAEMA-co-MMA)

DMAEMA

MMA

DMAEMA

MMA

Mn (g/mol)

PDI

PDM 1

6.36

9.98

40

60

56562

1.75

PDM 2

5.08

11.98

29

71

55865

1.65

PDM 3

3.82

13.98

20

80

54507

1.93

PDM 4

2.54

15.98

13

87

194617

1.88

Comment 9. The whole section 3.3. GPC results should be reconsidered. There is absolutely unclear statements: ”In synthesis of copolymer, monomers conversions resulted in a distribution of segment and formation of high molecular weight from bimolecular weighed distribution (MWD)”or “There was slight molecular weight distribution Mw/Mn, <1.4 of all copolymers [31].”

Reply: These all lines that suggested by reviewer were replaced (line 201-208). In synthesis of copolymer, monomers conversions resulted in high molecular weight copolymer that was same to the theoretical molecular weight [35]. Polydispersity index (PDI), Mw/Mn values characterize samples ranged from 1.65 to 1.93 for free radical polymerization [39]. The symmetrical nature of the GPC curves of four copolymers was referring the inexistence of an irreversible termination of both monomers, DMAEMA and MMA [36]. Traces of symmetric GPC distribution showed the uniformity of the copolymers via free radical polymerization. [31].These unimodal curves showed that the polymerization was completed successfully and that there was no unreacted monomers in the reaction product.

Comment 10. Section 3.4. Antibacterial Bioassay. Similarly, this section also contains unclear statements, for example “These polymethacrylamide random copolymers with different concentration of MMA and DMAEMA, have bearing protonated primary amine groups.”

Reply: These lines are replaced and now antibacterial bioassay has no more unclear statement.

Reviewer 3 Report

Reviewer 2:Comments and Suggestions for Mushtaq group

The manuscript ID polymers-970381 and titled as “Antibacterial Biofilm Adhesion Controlled Through Amphiphilic Copolymers of Dimethylamino Ethyl Methacrylate and Methyl Methacrylate” was reviewed. It is interesting paper. I accept the paper after some revision as indicated below.

  1. Abstract should be more quantitative. Some technical part should insert.
  2. The digital image of prepared copolymer should insert.
  3. Introduction needs some more recent papers by 2020. Introduce Recently reported similar work needs more comparison with present work.
  4. The zone inhibition presented in film is performed. But, it is quite shocked to see the film size containing random shape. Better all the sample should have circular and fix shape. Check and justify it.
  5. Conclusion should be technical.
  6. In abstract, the application related lines should mention.

Author Response

Reviewer 3 Comments

The manuscript ID polymers-970381 and titled as “Antibacterial Biofilm Adhesion Controlled through Amphiphilic Copolymers of Dimethylamino Ethyl Methacrylate and Methyl Methacrylate” was reviewed. It is interesting paper. I accept the paper after some revision as indicated below.

Comment 1. Abstract should be more quantitative. Some technical part should insert.

Reply: Abstract is more quantitative now here add technical part in lines (yellow highlighted).

Amphiphilic copolymers are recognized as important biomaterials and used as antibacterial agents due to their effective inhibition of bacterial growth. In current study the amphiphilic copolymers of P(DMAEMA-co-MMA) were synthesized using free radical polymerization by varying the concentrations of hydrophilic monomer 2-dimethylamino ethylmethacrylate (DMAEMA) and hydrophobic monomer methyl methacrylate (MMA) having PDI value of 1.65-1.93. The concentration of DMAEMA chain of copolymer serve to control the antibacterial ability of P(DMAEMA-co-MMA) to achieve optimized copolymer composition with high antibacterial activity. Antibacterial activity of synthesized copolymers was elucidated against Gram-positive Staphylococcus aureus (ATCC 6538) and Gram-negative Escherchia coli (ATCC 8739) by disk diffusion method and zones of inhibition were measured. The desired composition that was PDM1 copolymer had shown good zones of inhibition i.e. 19 ± 0.33 mm and 20 ± 0.33 mm for E. coli and S. aureus respectively. The PDM1 and PDM2 have exhibited significant control over bacterial biofilm adhesion as tested by six well plate method. SEM study of bacterial biofilm formation has illustrated that these copolymers act in a similar fashion like cationic biocide. These compositions viz. PDM1 and PDM2 may be useful in development of bioreactors, sensors, surgical equipment and drug delivery devices.

Comment 2. The digital image of prepared copolymer should insert.

Reply: Digital images of synthesized copolymers are inserted in manuscript as figure 2.

Comment 3.Introduction needs some more recent papers by 2020. Introduce recently reported similar work needs more comparison with present work.

Reply: Recent papers of 2020 are added in introduction and compared with the present work lines (73-81). Xu et al. investigated cationic polymers as the main classes of materials against bacteria through the membrane-lysis mechanism. They studied antibacterial effects of linear and cyclic monomers of cyclic poly(2-(dimethylamino)ethyl methacrylate) (PDMAEMA) based copolymers with various components by the intra-chain click cyclization of α-alkyne-ω-azido via atom transfer radical polymerization [26]. In another study of amphiphilic polymers of thiazole ring 2-(2-(4-methylthiazol-5-yl)ethoxy)carbonyl)oxy)ethyl methacrylate monomer (MTZ) and non-hemotoxic poly(ethylene glycol) side chains (poly(ethylene glycol) methyl ether methacrylate (PEGMA) observed that longer hydrophobic chain, octyl were much more hemotoxic than their corresponding butylated copolymers

Comment 4. The zone inhibition presented in film is performed. But, it is quite shocked to see the film size containing random shape. Better all the sample should have circular and fix shape. Check and justify it.

Reply: Zone of inhibition was performed for copolymers of fix shape and size. These amphiphilic copolymers are partially soluble in water so these look like irregular shape after 24 hours in agar medium. Otherwise these copolymers cut in regular shape and of fixed size of 8 mm.

Comment 5. Conclusion should be technical.

Reply: Conclusion is technically improved (lines 284-293).

Various copolymers (PDM1, PDM2, PDM3 and PDM4) were successfully prepared by free radical polymerization in presence of AIBN initiator by employing different ratios of DMAEMA and MMA monomers. These copolymers were characterized by FTIR, 1HNMR and GPC that exhibited high antimicrobial activities against E. coli and S. aureus whereas; bare PMMA showed no antibacterial activity. Copolymer of P(DMAEMA-co-MMA), PDM1 showed maximum biocidal activity with inhibition zone of 19 mm and 20 mm against E. coli and S. aureus respectively. In these copolymers PDM1 and PDM2 had higher concentration of DMAEMA therefore it showed greater antibacterial activities as compared to PDM3 and PDM4. This greater activity is attributed to the presence of amine groups along the chain length of DMAEMA segment. Adhesion of microorganisms on the surface and biofilm formation decreased with increase in molar ratio of DMAEMA due to presence of positive charges responsible for biocidal action. In SEM analysis of biofilm control and rupture of E. coli cell membrane was observed in PDM1 and cell disruption of S. aureus was observed in PDM2. Thus PDM1 and PDM2 copolymers are potential candidates for antifouling applications with controlled biofilm.

Comment 6. In abstract, the application related lines should mention.

Reply: Application related lines (28-30) are added in abstract.

These compositions viz. PDM1 and PDM2 may be useful in development of bioreactors, sensors, surgical equipment and drug delivery devices.

Reviewer 4 Report

This article presents good results, within a theme that has been extensively explored in the sciences of polymers and materials, in general, but needs some adjustments to be accepted. Here are some questions and suggestions.

1. Check the correct spelling of the bacteria.
2. S
tandardize numbers of decimal places in measurements.
3.
Tests for evaluating bacterial activity are well discussed in the literature, with concordance and disagreement of action. The direct contact test is the most reliable and has been increasingly used, as insoluble materials in the bacterial growth medium will never cause diffusion. This was shown in Illustration 1, that direct contact is the one that kills bacteria. Authors should rethink the tests used, explain whether and how the polymers used really diffuse and redo this illustration (which is indicating death by direct contact).
4. In the FTIR graph the allocation of the NH band for the PDMAEMA material is in a very different region from that shown in the Figure, it needs to be revised. The error must be occurring because there are no NH bands, since in the proposal in Figure 2, there is no NH, needing to re-identify this band.
5.
What happens with the C = O band of the PDMAEMA material?
6. Materials 1, 2 and 3 do not have differences in elution time in the GPC, however they presented different properties in relation to the tested bacteria, how to explain this?
7. The images for antibacterial activity need to be revised, as the diffusion is very complicated to observe and the authors need to make it clear how the calculation was done, since the diameter is calculated and no circumference was observed.
8.
The control needs to be shown in order to better compare the results.  

Author Response

Reviewer 4 Comments

Comment 1. Check the correct spelling of the bacteria.

Reply: spelling checked.

Comment 2. Standardize numbers of decimal places in measurements.

Reply: Standardized number of decimal in measurements.

Comment 3. Tests for evaluating bacterial activity are well discussed in the literature, with concordance and disagreement of action. The direct contact test is the most reliable and has been increasingly used, as insoluble materials in the bacterial growth medium will never cause diffusion. This was shown in Illustration 1, that direct contact is the one that kills bacteria. Authors should rethink the tests used, explain whether and how the polymers used really diffuse and redo this illustration (which is indicating death by direct contact).

Reply: Yes insoluble materials do not showed diffusion in bacteria like PMMA (poly methyl meth acrylate). In these copolymers of MMA (methyl methacrylate) and DMAEMA (Dimethyl amino ethyl meth acrylate) were used. Here MMA is water insoluble monomer and DMAEMA is water soluble and overall these copolymers are partially soluble in water. Basically disk diffusion method is used for those materials which are soluble or partially soluble in water. These copolymers P(DMAEMA-co-MMA) have different  content of hydrophilic and hydrophobic monomers that control the adhesion and rapturing of bacterial cell  wall and adhesion on the surface of copolymers. Previous reports of PDMAEMA against Gram-positive bacteria suggested that the mode of action involved membrane binding, followed by permeabilization of the cell membrane, enabling leakage of cytoplasmic contents. Accordingly, the quaternized P(DMAEMA) coating is effective against E. coli . There are hints that quaternary ammonium compounds have excellent bactericidal activities (Tiller et al. 2001; Gottenbos et al. 2002; Lin et al. 200 ; Cen et al. 2003 ). The antibacterial effect of quaternized P(DMAEMA) coating in the present study increased from 0 to 16 h, and its long-time bacteriostatic rate was more than 50 % . The antibacterial mechanism of special polymers involves the generation of very high local charge densities that are harmful to the central membrane functions of microorganisms. This biocidal effect is encompasses both Gram-positive and -negative bacteria, as well as fungi and yeasts. However, protonation of amino groups can further boost this effect and cell death occur.

References:

  1. Rawlinson, L.-A. B.; Ryan, S. M.; Mantovani, G.; Syrett, J. A.;Haddleton, D. M.; Brayden, D. J. Biomacromolecules 2010, 11 (2), 443−453.
  2. Tiller, J.C., Liao, C.J., Lewis, K., Klibanov, A.M. (2001) Designing surfaces that kill bacteria on contact. Proc. Natl. Acad. Sci. USA 98:5981 – 5985.
  3. Kandelbauer, A., Widsten, P. (2009) Antibacterial melamine resin surfaces for wood-based furniture and flooring. Prog. Org.Coat. 65:305 – 313.

Comment 4. In the FTIR graph the allocation of the NH band for the PDMAEMA material is in a very different region from that shown in the Figure, it needs to be revised. The error must be occurring because there are no NH bands, since in the proposal in Figure 2, there is no NH, needing to re-identify this band.

Reply:  FTIR is revised and the error in the Figure is removed and explain in lines (193-194)
 and now it justifies in manuscript and band in the range of 3000-3500 cm-1 in Figure 2 (a) was due to moisture absorbance because of hydrophilic nature of PDMAEMA.

Comment 5. What happens with the C = O band of the PDMAEMA material?

Reply:  In material C=O bond exist as shown in Figure 2a as well as in Figure 2b because in copolymerization no effect on C=O bond.

Comment 6. Materials 1, 2 and 3 do not have differences in elution time in the GPC, however they presented different properties in relation to the tested bacteria, how to explain this?

Reply:  These all materials have little difference in elution time and all copolymers are made up of acrylate monomers (DMAEMA and MMA). In copolymers 1, 2 and 3 ratio of DMAEMA is different that play key role in bacteriostatic action and control biofilm formation and adhesion of bacteria on the surface. More concentration of hydrophilic monomer (DMAEMA) had showed more antibacterial activity.

Reference: Kandelbauer, A., Widsten, P. (2009) Antibacterial melamine resin surfaces for wood-based furniture and flooring. Prog. Org.Coat. 65:305 – 313.

Comment 7. The images for antibacterial activity need to be revised, as the diffusion is very complicated to observe and the authors need to make it clear how the calculation was done, since the diameter is calculated and no circumference was observed.

Reply: In antibacterial study through disk diffusion method copolymers showed good zones of inhibition and these calculated very carefully. Calculation of zone of inhibition was done manually by measuring with scale after drawing circles around the inhibition zones. In this pandemic situation labs are not functional and performing again is not possible.

Comment 8. The control needs to be shown in order to better compare the results. 

Reply: In this research work control is PMMA that has no antibacterial activity and it’s already reported in our previous work.

Reference:Mushtaq, S.; Ahmad, N.M.; Nasir, H.; Mahmood, A.; Janjua, H.A. Transpicuous-Cum-Fouling Resistant Copolymers of 3-Sulfopropyl Methacrylate and Methyl Methacrylate for Optronics Applications in Aquatic Medium and Healthcare. 2020, 2020.

Round 2

Reviewer 1 Report

My previous comments are as follows: "The author should provide the NMR of PDM 2-4 for comparison." The authors should provide NMR spectra data with integration result in the figure not just list the result in the table.

Author Response

Comment1: My previous comments are as follows: "The author should provide the NMR of PDM 2-4 for comparison." The authors should provide NMR spectra data with integration result in the figure not just list the result in the table.

Reply: Requisite integrated 1HNMR spectra images captured from hard copies of PDM1 to PDM4 are appended below:-

(See attached file)

Reviewer 2 Report

The revised version displayed some amendments but still the manuscript cannot meet the criteria for publication.

The authors provided an extensive reply to the general comment that the reported results and the manuscript possess quite limited level of novelty. In their reply the authors referred to 12 different sources (references) rather than to discuss new and original results obtained in the investigation.

The authors supported the application of freeze-drying as a method for isolation and purification of the obtained products citing a publication by Schulz  et al. (D.N. Schulz, J.J. Kaladas, J.J. Maurer, J. Bock, S.J. Pace, W.W. Schulz, Copolymers of acrylamide and surfactant macromonomers: synthesis and solution properties, Polymer, Volume 28, Issue 12, 1987,Pages 2110-2115.). I read the publication and could say that Schulz et al. used dialysis as a purification method and they commented the effect of the isolation step. In addition the polymerisation systems are different and the intended applications also. Therefore, I can say it is not an appropriate reference. Anyway, when a biorelated application is the ultimate goal of a materials study the thorough purification of the material is needed.

The discussion and assignment of the IR spectra was improved however still there are contradicting statements. In the newly inserted text band (lines 184-185) the band at 1150 cm-1 is attributed to C-N stretching vibration while in their reply to reviewers’ comments the absorption in the region 1150-1250cm-1 is due to C-O-C deformation. Further, in the spectra of the copolymers the band at 1020 cm-1 attributed to C–N stretching is used as a proof for the presence of DMAEMA units in the copolymers.

In the subsection 3.2.1HNMR Analysis the authors has written that molar ratio of PDMAEMA and PMMA was 1:1 for PDM1 based on the NMR  data while the data reported in Table 1 the ratio is 2:3 (40:60 mol%).  Moreover, the ratio is between the DMAEMA and MMA units not between homopolymers.

A new comment was added in the subsection 3.3. GPC Results regarding the molecular weight of the copolymers that was close to the theoretical molecular weight [37]. The cited reference reports results on RAFT polymerisation which is a controlled process while the authors used in their investigation a free radical polymerisation.

The authors revised only the pointed examples by the reviewers’ and in the revised version there are still phrases that need amendment.

Just a couple of examples:

The caption of Figure 1: Schematic of copolymer synthesis and antibacterial potential of copolymers, P(DMAEMA and MMA)

In the Abstract, Line 20: The concentration of DMAEMA chain of copolymer serve ……

Also the authors offered keywords (Biofouling; Escherchia coli; Staphylococcus aureus; Optical density; Tryptosoy broth) that do not refer to the present investigation.

Based on the above I cannot recommend acceptance of the manuscript for publication.

Author Response

Comment 1: Anyway, when a biorelated application is the ultimate goal of a materials study the thorough purification of the material is needed.

Reply: In the free radical polymerization both monomers are polymerized using initiator and solvent. Here solvent is removed and no further purification is needed.

Comment 2: The discussion and assignment of the IR spectra was improved however still there are contradicting statements. In the newly inserted text band (lines 184-185) the band at 1150 cm-1 is attributed to C-N stretching vibration while in their reply to reviewers’ comments the absorption in the region 1150-1250cm-1 is due to C-O-C deformation. Further, in the spectra of the copolymers the band at 1020 cm-1 attributed to C–N stretching is used as a proof for the presence of DMAEMA units in the copolymers.

Reply:  In (Lines 185-186) both acrylate homopolymers band at 1150-1250 cm-1 showed stretching vibration of C-O-C [38]. On the other hand in copolymers band at 1020 cm-1 was due to C-N stretching vibration and the bands due to C-H vibrations of N(CH3)2 at 2842 cm-1 that showed the DMAEMA.

 Comment 3: In the subsection 3.2.1HNMR Analysis the authors has written that molar ratio of PDMAEMA and PMMA was 1:1 for PDM1 based on the NMR  data while the data reported in Table 1 the ratio is 2:3 (40:60 mol%).  Moreover, the ratio is between the DMAEMA and MMA units not between homopolymers.

Reply: 1HNMR analysis of PDM1 was performed and molar ratio of both monomers, DMAEMA and MMA in copolymer was calculated that 1:1 (44:56 mol%) (Lines 208-210) and it was added into the table. PDM1 spectra is given below.

Comment 4: A new comment was added in the subsection 3.3. GPC Results regarding the molecular weight of the copolymers that was close to the theoretical molecular weight [37]. The cited reference reports results on RAFT polymerisation which is a controlled process while the authors used in their investigation a free radical polymerisation.

Reply: Cited reference in GPC was replaced by free radical polymerization (Lines 220-221).

Comment 5: The authors revised only the pointed examples by the reviewers’ and in the revised version there are still phrases that need amendment.

Just a couple of examples:

The caption of Figure 1: Schematic of copolymer synthesis and antibacterial potential of copolymers, P(DMAEMA and MMA)

In the Abstract, Line 20: The concentration of DMAEMA chain of copolymer serve ……

Also the authors offered keywords (Biofouling; Escherchia coliStaphylococcus aureus; Optical density; Tryptosoy broth) that do not refer to the present investigation.

Reply: Some more amendments are done in manuscript that are pointed.

  • Schematic illustration of copolymers, P(DMAEMA and MMA) with varying concentration of DMAEMA and control bacterial adhesion. (Lines 97-98).
  • In Abstract line, 20: The DMAEMA monomer through ternary amine with antibacterial property optimized copolymers, P(DMAEMA-co-MMA) compositions to control biofilm adhesion.
  • Key Words : copolymerization; hydrophilic; antifouling; coli; S. aureus; biofilm (Line 31)
